# Incorporating Implicit Regularization to Enhance the Transition Matrix Method for Effective Handling of Diverse Label Noise

## Abstract

Among various methods for learning with noisy labels, the transition matrix method has attracted sustained attention due to its simplicity and statistical consistency. However, estimating the transition matrix for each sample may be unidentifiable and computationally expensive in the case of instance-dependent label noise and real-world situations. In this paper, we propose a concise method that only requires estimating a global matrix, combining with implicit regularization, to replace the estimation of the individual transition matrix for each sample. Specifically, by estimating the transition matrix, we can determine the overall probability transfer from correct labels to noisy labels and use implicit regularization to adjust the sparse form representation of the difference between the estimated posterior probability distribution and the noisy label distribution. This approach can be applied to diverse types of noise as well as alleviating the problem of inaccurate posterior probability estimation. We theoretically analyze the consistency and generalization results of the proposed method and conduct experiments on synthetic and real-world datasets with different types of label noise. The experimental results show that our method significantly outperforms previous transition matrix methods and has a wider range of applicability. Additionally, our method achieves impressive results without the need for additional auxiliary techniques. Our code will be open source and put on Github.

## 1 Introduction

Deep neural networks have achieved remarkable success in various fields in recent years, especially in classification problems with labeled data (Pouyanfar et al., 2018; Alom et al., 2019). Compared to traditional methods, deep neural networks have greatly improved performance but their effects heavily depend on the accuracy of the provided labels. Bringing data with corrupted labels into the neural network model without special treatment can severely affect the prediction performance (Daniely & Granot, 2019; Zhang et al., 2021a). However, acquiring accurately annotated data in reality can be very expensive, so a larger amount of data comes from the Internet or annotations by non-professional annotators. Therefore, it is currently worth studying and promoting how to alleviate the damage caused to the model when using noisy labels and make the model more robust, which is known as the problem of learning with noisy labels (Natarajan et al., 2013; Sukhbaatar et al., 2014; Han et al., 2018; Xia et al., 2019; 2020a; Algan & Ulusoy, 2021; Song et al., 2022).

Various methods have been proposed for learning with noisy labels. Existing methods can be classified into several categories. One of them is to design novel loss functions or network structures (Zhang & Sabuncu, 2018; Wang et al., 2019; Ma et al., 2020), which reduce the impact of noisy labels to make the model more robust. Another category is sample selection based on sample loss or feature extracted, dividing samples into the clean dataset and the noisy dataset (Arpit et al., 2017; Han et al., 2018; Jiang et al., 2018; Li et al., 2020). Then they relabel the noisy labels (Ren et al., 2018; Kremer et al., 2018), or clear the noisy labels and use semi-supervised methods for learning (Arazo et al., 2019; Li et al., 2020). These methods are common recently and have achieved some good results. However, the process of sample selection is relatively subjective, and statistical consistency is lost after the selection, and most of them lack theoretical support. In contrast, transition matrix methods (Goldberger & Ben-Reuven, 2016; Xia et al., 2019; Li et al., 2021b; Jiang

et al., 2021; Zhu et al., 2022) have statistical consistency and usually have corresponding theoretical analysis as support, attracting continued attention and occupying an important position in various learning algorithms with noisy labels.

The core idea of transition matrix methods is to use a matrix $\boldsymbol{T}(\boldsymbol{x})$ measuring the transition probability from the distribution of true label $P(\boldsymbol{Y}|X = \boldsymbol{x})$ to the distribution of observed noisy label $P(\tilde{\boldsymbol{Y}}|X = \boldsymbol{x})$ for given sample $X = \boldsymbol{x}$. If an accurate transition matrix can be estimated and combined with observable data to obtain the noisy class-posterior probability $P(\tilde{\boldsymbol{Y}}|X = \boldsymbol{x})$, the distribution of clean label $P(\boldsymbol{Y}|X = \boldsymbol{x})$ can be inferred for network learning. Therefore, estimating the transition matrix is the key to this type of method. However, it is infeasible to estimate an individual transition matrix for each sample without additional conditions (Liu et al., 2023). Previous methods mostly focus on class-dependent and instance-independent label noise problems (Xia et al., 2019; Li et al., 2021b; Zhang et al., 2021b), assuming that the transition matrix is fixed for all samples, i.e., $\boldsymbol{T}(\boldsymbol{x}) \equiv \boldsymbol{T}$. Even in this case, additional assumptions are still required. Some methods (Patrini et al., 2017; Xia et al., 2019) assume the existence of anchor points to estimate the transition matrix, while other methods obtain the optimal estimation by adding a regularization term for matrix structure to weaken the anchor points assumption (Li et al., 2021b; Zhang et al., 2021b). However, these methods are not suitable for instance-dependent label noise and complex real-world data because they estimate only one matrix for all samples. Moreover, when the estimation of noisy class-posterior distribution is inaccurate, the estimation of the transition matrix may be easily affected (Yao et al., 2020), thereby affecting the estimation of the clean label distribution. Although some new methods (Xia et al., 2020b; Zhu et al., 2021; Zhang & Sugiyama, 2021; Li et al., 2022) have recently been designed to use special networks or structures for instance-dependent noise situations, the estimation errors for them are still large, and the computational cost is too high to lose the concise characteristic of transition matrix methods.

To address the existing shortcomings of transition matrix methods, in this paper, we propose a method that only requires estimating a global transition matrix $\boldsymbol{T}$ applicable to various types of noise. We use this matrix to estimate the overall transfer trend of the real labels to noisy labels and then attempt to measure the difference between the transfer posterior probability $\boldsymbol{T}^{\top} P(\boldsymbol{Y}|X)$ and the noisy class-posterior probability distribution $P(\tilde{\boldsymbol{Y}}|X)$. When a suitable global matrix is applied, this difference should be relatively small. We use implicit regularization (Patrini et al., 2017; Liu et al., 2022) to model the sparsity of this residual term $P(\tilde{\boldsymbol{Y}}|X) - \boldsymbol{T}^{\top} P(\boldsymbol{Y}|X)$, and directly utilize the gradient method to update model parameters. Compared to traditional transition matrix methods for class-dependent label noise, our method does not require much additional time consumption. Compared to methods estimating transition matrices for each sample, it greatly reduces the computational time and space consumption by reducing the parameters to be estimated. In addition, for the problem of inaccurate noisy class-posterior estimation, our model can effectively mitigate its negative impact through the handling of fitting the residual term.

The structure of the following sections is as follows. In Section 2, we give relevant definitions and propose our method and corresponding algorithm. In section 3 we conduct a theoretical analysis of the proposed method on a simplified model. In Section 4, we conduct experiments on various synthetic and real-world noisy datasets, comparing with other methods. We conclude the paper in Section 5. In addition, we provide a more specific review of related works in Appendix A, algorithm framework and proofs of theorems in Appendix B, experimental details in Appendix C.

The main contributions of this paper are:

- We propose a novel and concise transition matrix method, which combines with implicit regularization, to effectively handle learning with various types of label noise data by only estimating a global matrix.

- Under certain assumptions, we provide theoretical analysis on the consistency and generalization results of the algorithm on a simplified model. We prove the theorems proposed accordingly, giving support for the effectiveness of the proposed method.

- Our proposed method achieves significant improvements compared to previous transition matrix methods on both synthetic and real-world noisy label datasets, and produces competitive results without the need for additional auxiliary techniques.

## 2 METHODOLOGY

In this section, based on previous transition matrix methods, we propose a novel approach that combines the transition matrix with implicit regularization (TMR) for learning labels. It is a convenient and end-to-end model that does not rely on assumptions about specific types of label noise. We will formulate the method in detail and illustrate it theoretically.

### 2.1 PRELIMINARIES

Let $\mathcal{X} \subset \mathbb{R}^d$ be the feature space, $\mathcal{Y} = \{1, 2, \cdots, C\}$ be the label space, where $C$ is the number of classes. Random variables $(X, Y), (X, \tilde{Y}) \in \mathcal{X} \times \mathcal{Y}$ denote the underlying data distributions with true and noisy labels respectively. In general, we can not observe the latent true data samples $\mathbb{D}_{(N)} = \{(\boldsymbol{x}_i, y_i)\}_{i=1}^N$, but can only obtain the corrupted data $\tilde{\mathbb{D}}_{(N)} = \{(\boldsymbol{x}_i, \tilde{y}_i)\}_{i=1}^N$, where $\tilde{y} \in \mathcal{Y}$ is the noisy label corrupted from the true label $y$, while denote corresponding one-hot label as $\boldsymbol{y}$ and $\tilde{\boldsymbol{y}}$.

Transition matrix methods use a matrix $\boldsymbol{T}(\boldsymbol{x}) \in [0, 1]^{C \times C}$ to represent the probability from clean label to noisy label, where the $ij$-th entry of the transition matrix is the probability that the instance $\boldsymbol{x}$ with the clean label $i$ corrupted to a noisy label $j$. The matrix satisfies the requirement that the sum of each row $\sum_{j=1}^C \boldsymbol{T}_{ij}(\boldsymbol{x})$ is 1, and usually has the requirement for diagonally dominant, i.e., $\boldsymbol{T}_{ii}(\boldsymbol{x}) > \boldsymbol{T}_{ij}(\boldsymbol{x}), \forall j \neq i$. Let $P(\boldsymbol{Y}|X = \boldsymbol{x}) = [P(Y = 1|X = \boldsymbol{x}), \cdots, P(Y = C|X = \boldsymbol{x})]^\top$ be the clean class-posterior probability and $P(\tilde{\boldsymbol{Y}}|X = \boldsymbol{x}) = [P(\tilde{Y} = 1|X = \boldsymbol{x}), \cdots, P(\tilde{Y} = C|X = \boldsymbol{x})]^\top$ be the noisy class-posterior probability, the formula can be write as:

$$P(\tilde{\boldsymbol{Y}}|X = \boldsymbol{x}) = \boldsymbol{T}(\boldsymbol{x})^\top P(\boldsymbol{Y}|X = \boldsymbol{x}). \tag{1}$$

Though estimating the transition matrix and the noisy class-posterior probability, the clean class-posterior probability can be inferred by $P(\boldsymbol{Y}|X = \boldsymbol{x}) = \boldsymbol{T}(\boldsymbol{x})^{-\top} P(\tilde{\boldsymbol{Y}}|X = \boldsymbol{x})$. Since it is difficult to estimate the transition matrix $\boldsymbol{T}(\boldsymbol{x})$ individually for each sample, the majority of existing methods (Patrini et al., 2017; Han et al., 2018; Li et al., 2021b) focus on studying the class-dependent and instance-independent transition matrix, i.e., $\boldsymbol{T}(\boldsymbol{x}) = \boldsymbol{T}$ for $\forall \boldsymbol{x}$. Although under such conditions, the transition matrix is still unidentifiable without any additional assumption, due to there are different $\boldsymbol{T}$ and $P(\boldsymbol{Y}|X = \boldsymbol{x})$ such that $P(\tilde{\boldsymbol{Y}}|X = \boldsymbol{x}) = \boldsymbol{T}_1^\top P_1(\boldsymbol{Y}|X = \boldsymbol{x}) = \boldsymbol{T}_2^\top P_2(\boldsymbol{Y}|X = \boldsymbol{x})$.

To solve this problem, some algorithms (Liu & Tao, 2015; Xia et al., 2019) assume that there exist anchor points for each class, i.e., there exists an instance $\mathrm{x}^i \in \mathcal{X}$ such that $P(Y = i|X = \boldsymbol{x}^i) = 1$ for $\forall i \in \{1, 2, \cdots, C\}$. Then they can estimate the transition matrix by:

$$P\left(\tilde{Y} = j \mid X = \boldsymbol{x}^i\right) = \sum_{k=1}^C T_{kj} P\left(Y = k \mid X = \boldsymbol{x}^i\right) = T_{ij}. \tag{2}$$

However, the assumption of anchor points is not always valid, and some methods (Li et al., 2021b; Zhang et al., 2021b) have been proposed to weaken this assumption through special designs. Among them, Li et al. (2021b) tries to make the transition matrix identifiable by solving the optimization problem:

$$\min_{\boldsymbol{T} \in \mathbb{T}} \mathrm{Vol}(\boldsymbol{T}) \tag{3}$$
$$\text{s.t. } \boldsymbol{T}^\top f_{\boldsymbol{\theta}}(X) = P(\tilde{\boldsymbol{Y}}|X),$$

where $f_{\boldsymbol{\theta}} : \mathcal{X} \to \Delta^{C-1}$ ($\Delta^{C-1} \subset [0, 1]^C$ is the $C$-dimensional simplex) is a differentiable function represented by a neural network with parameters $\boldsymbol{\theta}$ to learn the clean class-posterior $P(\boldsymbol{Y}|X = \boldsymbol{x})$. $\mathbb{T} = \left\{\boldsymbol{T} \in [0, 1]^{C \times C} | \sum_{j=1}^C \boldsymbol{T}_{ij} = 1, \boldsymbol{T}_{ii} > \boldsymbol{T}_{ij}, \forall j \neq i\right\}$ is the set of diagonally dominant transition matrices. $\mathrm{Vol}(\boldsymbol{T})$ denotes a measure that is related to the volume of the simplex formed by the columns of $\boldsymbol{T}$, such as $\mathrm{Vol}(\boldsymbol{T}) = \det(\boldsymbol{T})$ or $\log \det(\boldsymbol{T})$.

According to the KKT condition (Karush, 1939) and adopt $\mathrm{Vol}(\boldsymbol{T}) = \log \det(\boldsymbol{T})$, the optimization problem 3 become to optimize the following loss function:

$$\mathcal{L}_1(\boldsymbol{\theta}, \boldsymbol{T}) = \frac{1}{N} \sum_{i=1}^N \ell\left(\boldsymbol{T}^\top f_{\boldsymbol{\theta}}\left(\boldsymbol{x}_i\right), \tilde{\boldsymbol{y}}_i\right) + \lambda \cdot \log \det(\boldsymbol{T}), \tag{4}$$

where $\ell$ is a loss function usually using cross-entropy(CE) loss, $\tilde{\boldsymbol{y}}_i$ is the one-hot label corresponding to $\tilde{y}_i$ and $\lambda > 0$ is a regularization coefficient that balances CE loss versus volume minimization. This volume minimization transition matrix method is denoted by VolMinNet and it is an end-to-end framework without the need for identifying anchor points or a second stage for loss correction.

However, this method is limited by the assumption of class-dependence and cannot be applied to data with instance-dependent label noise. Additionally, its effectiveness closely relies on the accurate estimation of the noisy class-posterior probability. When the estimation is inaccurate, the algorithm's performance will be damaged (Yao et al., 2020). Following this method, we improve it by combining it with implicit regularization to form a new approach called TMR.

## 2.2 Transition Matrix with Implicit Regularization

As mentioned in the previous section, although methods like VolMinNet ensure the transition matrix identifiable, it is not applicable to situations with instance-dependent label noise and the performance is heavily influenced by the accuracy of the noisy class-posterior probability estimation. The main reason for this is that the product of the transition matrix $\boldsymbol{T}$ and clean class-posterior probability $P(\boldsymbol{Y}|X)$, i.e., $\boldsymbol{T}^\top P(\boldsymbol{Y}|X)$ is not always equal to the noisy class-posterior probability $P(\tilde{\boldsymbol{Y}}|X)$. For instance-dependent label noise, it is not enough to make the equation 1 hold for any $X$. While for class-dependent label noise data, it is difficult to obtain an accurate estimation of the noisy class-posterior probability through the randomness of limited training data, so the estimation of the transition matrix and clean label distribution based on the second equation of 3 is poorly estimated.

We use a residual vector $\boldsymbol{\gamma}(X)$ with respect to feature $X$ to measure the distribution difference between $P(\tilde{\boldsymbol{Y}}|X)$ and $\boldsymbol{T}^\top P(\boldsymbol{Y}|X)$, i.e., as follow:

$$\boldsymbol{T}^\top P(\boldsymbol{Y}|X) + \boldsymbol{\gamma}(X) = P(\tilde{\boldsymbol{Y}}|X). \tag{5}$$

As can be seen from the above formula 5, if a valid transition matrix $\boldsymbol{T}$ and residual term $\boldsymbol{\gamma}(X)$ can be estimated, then a clean class-posterior probability can be obtained, regardless of instance-dependent noise or the noisy class-posterior has estimation error. Therefore, the core of our proposed method lies in using an overall transition matrix and sample residual term to replace the estimation of a separate transition matrix for each sample. In this way, the number of parameters for the transition matrix is reduced from $O(NC^2)$ to $O(NC)$, which greatly reduces the difficulty of matrix estimation and computational consumption when $C$ is large. However, it is still unrealistic to estimate a residual for each sample without other constraints, so additional model assumptions need to be added to make the problem solvable.

Intuitively, if an overall relatively suitable transition matrix is applied to $\boldsymbol{T}^\top P(\boldsymbol{Y}|X)$, then the difference between it and the probability $P(\tilde{\boldsymbol{Y}}|X)$ should be small. Therefore, in our work on probabilistic modeling, we set the difference $\boldsymbol{\gamma}$ to be sparse for the training data. Inspired by using implicit regularization to represent sparse structures (Neyshabur et al., 2014; Patrini et al., 2017; Liu et al., 2022), we exploit this technique to estimate $\boldsymbol{\gamma}_i$ as $\boldsymbol{r}_i = \boldsymbol{u}_i \odot \boldsymbol{u}_i - \boldsymbol{v}_i \odot \boldsymbol{v}_i$ with respect to training sample $\boldsymbol{x}_i$, where $\odot$ denotes an entry-wise Hadamard product. As usual, we use a deep neural network $f_{\boldsymbol{\theta}}(\cdot)$ to learn the true label probability $\boldsymbol{y}_i$ w.r.t $\boldsymbol{x}_i$. So for the noisy label probability distribution $\tilde{\boldsymbol{y}}_i$ given by the data, the model use $\boldsymbol{T}^\top f_{\boldsymbol{\theta}}(\boldsymbol{x}_i) + \boldsymbol{u}_i \odot \boldsymbol{u}_i - \boldsymbol{v}_i \odot \boldsymbol{v}_i$ to fit it. Bring it into the loss function as:

$$\mathcal{L}_2(\boldsymbol{\theta}, \boldsymbol{T}, \{\boldsymbol{u}_i, \boldsymbol{v}_i\}_{i=1}^N) = \frac{1}{N}\sum_{i=1}^N \ell\left(\boldsymbol{T}^\top f_{\boldsymbol{\theta}}(\boldsymbol{x}_i) + \boldsymbol{u}_i \odot \boldsymbol{u}_i - \boldsymbol{v}_i \odot \boldsymbol{v}_i, \tilde{\boldsymbol{y}}_i\right). \tag{6}$$

To ensure the transition matrix is identifiable, we add a regularization term of the volume of the matrix to loss function as Li et al. (2021b) using. The total loss function applied in our proposed method is:

$$\mathcal{L}(\boldsymbol{\theta}, \boldsymbol{T}, \{\boldsymbol{u}_i, \boldsymbol{v}_i\}_{i=1}^N) = \frac{1}{N}\sum_{i=1}^N \ell\left(\boldsymbol{T}^\top f_{\boldsymbol{\theta}}(\boldsymbol{x}_i) + \boldsymbol{u}_i \odot \boldsymbol{u}_i - \boldsymbol{v}_i \odot \boldsymbol{v}_i, \tilde{\boldsymbol{y}}_i\right) + \lambda \cdot \log \det(\boldsymbol{T}), \tag{7}$$

where we estimate parameters according to:

$$\hat{\boldsymbol{\theta}}, \hat{\boldsymbol{T}}, \{\hat{\boldsymbol{u}}_i, \hat{\boldsymbol{v}}_i\}_{i=1}^N = \underset{\boldsymbol{\theta}, \boldsymbol{T}, \{\boldsymbol{u}_i, \boldsymbol{v}_i\}_{i=1}^N}{\arg\min} \mathcal{L}(\boldsymbol{\theta}, \boldsymbol{T}, \{\boldsymbol{u}_i, \boldsymbol{v}_i\}_{i=1}^N). \tag{8}$$

We use the gradient descent method to update the parameters to be learned above. The method steps are summarized in Algorithm 1 in Appendix B.1.

## 3 THEORETICAL ANALYSIS

In this section, we want to analyze the effectiveness of the proposed method theoretically. However, it is difficult to give a direct analysis of the deep neural network model. So we follow the theoretical analysis method of Liu et al. (2022) to simplify the proposed model and study on an approximately linear structure.

### 3.1 MODEL SIMPLIFICATION AND CONSISTENCY ANALYSIS

The first to solve is the construction of an approximate simplified model for theoretical analysis of our algorithm. Based on Jacot et al. (2018), we use first-order Taylor expansion to approximate the deep neural network $f_{\boldsymbol{\theta}}(\cdot)$, which is highly over-parameterized:

$$f_{\boldsymbol{\theta}}(\boldsymbol{x}) \approx f_{\boldsymbol{\theta}_0}(\boldsymbol{x}) + \left( \frac{\partial f_{\boldsymbol{\theta}}^{\top}(\boldsymbol{x})}{\partial \boldsymbol{\theta}} \Big|_{\boldsymbol{\theta}=\boldsymbol{\theta}_0} \right)^{\top} \cdot (\boldsymbol{\theta} - \boldsymbol{\theta}_0), \tag{9}$$

where $f_{\boldsymbol{\theta}}(\boldsymbol{x})$ is a C-dimensional vector, $\boldsymbol{\theta} \in \mathbb{R}^p$ $(p \gg N)$ denotes the parameters of the neural network, $\frac{\partial f_{\boldsymbol{\theta}}^{\top}(\boldsymbol{x})}{\partial \boldsymbol{\theta}} \Big|_{\boldsymbol{\theta}=\boldsymbol{\theta}_0}$ is a $p \times C$ matrix, $\boldsymbol{\theta}_0$ is the initialization of $\boldsymbol{\theta}$, symbol $\cdot$ represents matrix multiplication. For simplicity, we drop the constant term in the derivation and abbreviate $\frac{\partial f_{\boldsymbol{\theta}}^{\top}(\boldsymbol{x})}{\partial \boldsymbol{\theta}} \Big|_{\boldsymbol{\theta}=\boldsymbol{\theta}_0}$ as $\nabla_{\boldsymbol{\theta}_0} f(\boldsymbol{x})$. The approximate formula becomes:

$$f_{\boldsymbol{\theta}}(\boldsymbol{x}) \approx \nabla_{\boldsymbol{\theta}_0} f(\boldsymbol{x})^{\top} \cdot \boldsymbol{\theta}. \tag{10}$$

Through this processing, we simplify the deep neural network into an approximately linear structure, and we use $f_{\boldsymbol{\theta}}(\boldsymbol{x}) = \nabla_{\boldsymbol{\theta}_0} f(\boldsymbol{x}) \cdot \boldsymbol{\theta}$ in the following theoretical analysis. We use a $N \times C$ matrix $\boldsymbol{F}$ to represent the neural network predictions on the overall training dataset $\{(\boldsymbol{x}_i, y_i)\}_{i=1}^{N}$:

$$\boldsymbol{F} = \begin{bmatrix} f_{\boldsymbol{\theta}}^{\top}(\boldsymbol{x}_1) \\ \vdots \\ f_{\boldsymbol{\theta}}^{\top}(\boldsymbol{x}_N) \end{bmatrix}. \tag{11}$$

In order to be written in matrix form, we rewrite the formula 10 in vector expansion form:

$$f_{\boldsymbol{\theta}}^{\top}(\boldsymbol{x}) = [f_{\boldsymbol{\theta}}(\boldsymbol{x})_1, \cdots, f_{\boldsymbol{\theta}}(\boldsymbol{x})_C] = \text{vec}(\nabla_{\boldsymbol{\theta}_0} f(\boldsymbol{x}))^{\top} \cdot \Theta, \tag{12}$$

where $\text{vec}(\boldsymbol{A})$ denotes matrix expansion of a $m \times n$ matrix $\boldsymbol{A}$ by column vectors:

$$\text{vec}(\boldsymbol{A}) = [\boldsymbol{A}_{1,1}, \cdots, \boldsymbol{A}_{m,1}, \cdots, \boldsymbol{A}_{1,n}, \cdots, \boldsymbol{A}_{m,n}]^{\top}, \tag{13}$$

and $\Theta$ is a $CP \times C$ matrix, denoting the Kronecker product of $C \times C$ identity matrix $\boldsymbol{I}_C$ with $\boldsymbol{\theta}$, i.e.,

$$\Theta = \boldsymbol{I}_C \otimes \boldsymbol{\theta} = \begin{bmatrix} \boldsymbol{\theta} & 0 & \cdots & 0 \\ 0 & \boldsymbol{\theta} & \cdots & 0 \\ \vdots & \vdots & \ddots & \vdots \\ 0 & 0 & \cdots & \boldsymbol{\theta} \end{bmatrix}_{CP \times C}. \tag{14}$$

We use a Jacobian matrix $\boldsymbol{G} \in \mathbb{R}^{N \times CP}$ to denote the partial derivatives of the network for each sample:

$$\boldsymbol{G} = \begin{bmatrix} \text{vec}(\nabla_{\boldsymbol{\theta}_0} f(\boldsymbol{x}_1))^{\top} \\ \vdots \\ \text{vec}(\nabla_{\boldsymbol{\theta}_0} f(\boldsymbol{x}_N))^{\top} \end{bmatrix}. \tag{15}$$

Then, an aggregate form of 10 is:

$$\boldsymbol{F} = \boldsymbol{G} \cdot \Theta. \tag{16}$$

Now we give a simplified model assumption that there exists an underlying ground truth parameter $\boldsymbol{\theta}_*$ such that corresponding $\boldsymbol{F}_*$ generated by 16 fits the true label distribution for sample. Meanwhile, there exist potentially true transition matrix $\boldsymbol{T}_*$ and sparse residual matrix $\boldsymbol{R}_* =$

$[\boldsymbol{\gamma}(\boldsymbol{x}_1), \cdots, \boldsymbol{\gamma}(\boldsymbol{x}_N)]^\top$ made up of the residual terms $\boldsymbol{\gamma}(\boldsymbol{x})$ for sample defined in Section 2.2. We assume that the $N \times C$ observed noisy label matrix $\tilde{\boldsymbol{Y}} = [\tilde{\boldsymbol{y}}_1, \cdots, \tilde{\boldsymbol{y}}_N]^\top$ is generated by:

$$\tilde{\boldsymbol{Y}} = \boldsymbol{F}_* \cdot \boldsymbol{T}_* + \boldsymbol{R}_*. \tag{17}$$

Expanded form after bringing in $\boldsymbol{G}$ and $\boldsymbol{\theta}_*$ is:

$$\tilde{\boldsymbol{Y}} = \boldsymbol{G} \cdot (\boldsymbol{I}_C \otimes \boldsymbol{\theta}_*) \cdot \boldsymbol{T}_* + \boldsymbol{R}_*. \tag{18}$$

The problem to be studied is transformed into given $\boldsymbol{G}$ and observed $\tilde{\boldsymbol{Y}}$ generated by 18, how to estimate the underlying $\boldsymbol{\theta}_*$, $\boldsymbol{T}_*$ and $\boldsymbol{R}_*$. At this time, our proposed loss function to be optimized 7 transforms into:

$$\mathcal{L}(\boldsymbol{\theta}, \boldsymbol{T}, \boldsymbol{U}, \boldsymbol{V}) = L\left(\boldsymbol{G} \cdot (\boldsymbol{I}_C \otimes \boldsymbol{\theta}) \cdot \boldsymbol{T} + \boldsymbol{U} \odot \boldsymbol{U} - \boldsymbol{V} \odot \boldsymbol{V}, \tilde{\boldsymbol{Y}}\right) + \lambda \cdot \log \det(\boldsymbol{T}), \tag{19}$$

where $L$ is matrix form from $\ell$ in 7, $\boldsymbol{U} = [\boldsymbol{u}_1, \cdots, \boldsymbol{u}_N]^\top$, $\boldsymbol{V} = [\boldsymbol{v}_1, \cdots, \boldsymbol{v}_N]^\top$, $\boldsymbol{R} = \boldsymbol{U} \odot \boldsymbol{U} - \boldsymbol{V} \odot \boldsymbol{V}$.

Intuitively, the parameters $\boldsymbol{\theta}, \boldsymbol{T}, \boldsymbol{R}$ are unidentifiable without other conditions due to the model 18 is over-parameterized. We need to add some conditional assumptions to ensure the convergence consistency of parameters. The required conditions are summarized in the Appendix B.2, such as the low rank condition of $\boldsymbol{G}$, sparsity of $\boldsymbol{R}_*$, special small initialization setting, sufficiently scattered assumption (Li et al., 2021b) of clean class-posterior probability distribution, etc. Under these conditions, we try to analyze the effectiveness of our algorithm. For the simplicity of proof, we use square loss in 19, which can be analogized to cross-entropy loss. The parameter optimization problem 8 becomes:

$$\hat{\boldsymbol{\theta}}, \hat{\boldsymbol{T}}, \hat{\boldsymbol{U}}, \hat{\boldsymbol{V}} = \underset{\boldsymbol{\theta}, \boldsymbol{T}, \boldsymbol{U}, \boldsymbol{V}}{\arg \min} \frac{1}{2} \|\boldsymbol{G} \cdot (\boldsymbol{I}_C \otimes \boldsymbol{\theta}) \cdot \boldsymbol{T} + \boldsymbol{U} \odot \boldsymbol{U} - \boldsymbol{V} \odot \boldsymbol{V} - \tilde{\boldsymbol{Y}}\|_2^2 + \lambda \cdot \log \det(\boldsymbol{T}). \tag{20}$$

Based on this, the consistency result of parameters estimation is as follows:

**Theorem 1.** (**Consistency**) *Under the conditions in B.2, the estimated parameters $\hat{\boldsymbol{\theta}}$, $\hat{\boldsymbol{T}}$, $\hat{\boldsymbol{R}}$ for optimization problem 20 based on algorithm 1 converge to the ground truth solution $\boldsymbol{\theta}_*$, $\boldsymbol{T}_*$, $\boldsymbol{R}_*$.*

The proof can be seen in Appendix B.3. Theorem 1 shows that under a simplified linear model and some conditions, one can use our proposed algorithm to obtain the consistent estimation of network parameters $\boldsymbol{\theta}_*$ applicable to learning with clean label data. At the same time, we can estimate the overall transition probability $\boldsymbol{T}_*$ from the correct label to the noisy label that we observed. Theorem 1 provides theoretical support for the effectiveness of our proposed method.

## 3.2 GENERALIZATION ANALYSIS

In addition to consistency, the generalization of the proposed result is also worth exploring. It is finite to the amount of noisy label training data $\tilde{\mathbb{D}}_{(N)} = \{(\boldsymbol{x}_i, \tilde{y}_i)\}_{i=1}^N$ we can observe, which is considered to be randomly sampled from the overall infinite noisy data $\tilde{\mathbb{D}}$. We want to explore how well the parameters $\hat{\boldsymbol{\theta}}_{(N)}, \hat{\boldsymbol{T}}_{(N)}$ estimated by the proposed algorithm with finite data $\tilde{\mathbb{D}}_{(N)}$ fit when applied to the overall data $\tilde{\mathbb{D}}$.

Define the function class be $\mathcal{F} := \left\{\ell(\boldsymbol{T}^\top f_{\boldsymbol{\theta}}(\cdot) + \boldsymbol{\gamma}(\cdot), \cdot) : \mathcal{X} \times \mathcal{Y} \to \mathbb{R}^+, \forall \boldsymbol{\theta} \in \mathbb{R}^p, \boldsymbol{T} \in \mathbb{T}\right\}$, where $\boldsymbol{\gamma}(\cdot)$ is the true residual term for each sample. Each element in $\mathcal{F}$ is a function about data sample. It is worth mentioning that the term of $\log \det(\boldsymbol{T})$ can be ignored in this part of analysis and does not affect the results. Denote the $\epsilon$-cover of $\mathcal{F}$ as $\mathcal{N}_{\mathcal{F}} = \mathcal{N}(\epsilon, \mathcal{F}, \|\cdot\|_\infty)$, the average losses on $\tilde{\mathbb{D}}_{(N)}$ and $\tilde{\mathbb{D}}$ are $\mathcal{L}(\boldsymbol{\theta}_{(N)}, \boldsymbol{T}_{(N)}, \boldsymbol{R}_{(N)}; \tilde{\mathbb{D}}_{(N)})$ and $\mathcal{L}(\boldsymbol{\theta}, \boldsymbol{T}, \boldsymbol{R}; \tilde{\mathbb{D}})$ respectively. According to Theorem 1, for any fixed $\epsilon > 0$, there exists estimated parameters $\hat{\boldsymbol{\theta}}_{(N)}, \hat{\boldsymbol{T}}_{(N)}, \hat{\boldsymbol{R}}_{(N)}$ obtained by our algorithm such that:

$$\mathcal{L}(\hat{\boldsymbol{\theta}}_{(N)}, \hat{\boldsymbol{T}}_{(N)}, \hat{\boldsymbol{R}}_{(N)}; \tilde{\mathbb{D}}_{(N)}) \le \mathcal{L}(\boldsymbol{\theta}_{(N)}, \boldsymbol{T}_{(N)}, \boldsymbol{R}^*_{(N)}; \tilde{\mathbb{D}}_{(N)}), \forall \boldsymbol{\theta}_{(N)} \in \mathbb{R}^p, \boldsymbol{T}_{(N)} \in \mathbb{T} \tag{21}$$

where $\boldsymbol{R}^*_{(N)}$ is the true residual terms for $\tilde{\mathbb{D}}_{(N)}$. If we know the ground truth $\boldsymbol{R}_*$, we have the following result:

**Theorem 2.** *Suppose the loss function is bounded by $0 \leq \ell(\cdot, \cdot) \leq M$. For any $\delta > 0$, then with probability at least $1 - \delta$ we have*

$$\mathcal{L}(\hat{\boldsymbol{\theta}}_{(N)}, \hat{\boldsymbol{T}}_{(N)}, \boldsymbol{R}_*; \tilde{\mathbb{D}}) \leq \inf_{\boldsymbol{\theta} \in \mathbb{R}^p, \boldsymbol{T} \in \mathbb{T}} \mathcal{L}(\boldsymbol{\theta}, \boldsymbol{T}, \boldsymbol{R}^*; \tilde{\mathbb{D}}) + M\sqrt{\frac{\ln(2\mathcal{N}_{\mathcal{F}}/\delta)}{2n}} + M\sqrt{\frac{\ln(2/\delta)}{2n}} + 3\epsilon. \quad (22)$$

The proof can be found in Appendix B.4, using Theorem 2 in Yong et al. (2022) as a reference. Looking back at the optimization target 20, we can find that the Theorem 2 states the estimators $\hat{\boldsymbol{\theta}}_{(N)}, \hat{\boldsymbol{T}}_{(N)}$ based on finite data $\tilde{\mathbb{D}}_{(N)}$ can also be applied relatively effectively to wider data $\tilde{\mathbb{D}}$ as long as they are randomly generated from the same pattern. It shows the generalization result of our algorithm, indicating that the estimation $\hat{\boldsymbol{\theta}}_{(N)}, \hat{\boldsymbol{T}}_{(N)}$ can be applied to new data and only the residual terms $\boldsymbol{R}$ need to be estimated separately.

Table 1: Test accuracy with symmetric and flip noise on CIFAR-10/100.

| | CIFAR-10 | | | |
| --- | --- | --- | --- | --- |
| | Symmetric | | Flip | |
| | 20% | 50% | 20% | 45% |
| GCE | 87.83±0.54 | 79.54±0.23 | 89.75±1.53 | 75.75±0.36 |
| Forward | 85.20±0.80 | 74.82±0.78 | 88.21±0.48 | 77.44±6.89 |
| Co-teaching | 82.27±0.07 | 75.55±0.07 | 80.65±0.20 | 73.02 ±0.23 |
| DMI | 87.54±0.20 | 82.68±0.21 | 89.89±0.45 | 73.15±7.31 |
| T-Revision | 87.95±0.36 | 80.01±0.62 | 90.33±0.52 | 78.94±2.58 |
| Dual T | 88.35±0.33 | 82.54±0.19 | 89.77 ±0.25 | 76.53 ±2.51 |
| TVD | 88.89±0.21 | 83.21±0.13 | 90.01±0.23 | 88.15±0.10 |
| VolMinNet | 89.58±0.26 | 83.37±0.25 | 90.37±0.30 | 88.54±0.21 |
| ROBOT | 92.13±0.07 | 88.75±0.10 | 93.70±0.07 | 92.52±0.16 |
| SOP | 93.18±0.57 | 88.98±0.43 | 94.02±0.30 | 89.58±0.86 |
| TMR | **94.17±0.22** | **91.36±0.31** | **94.45±0.19** | **93.01±0.62** |
| | CIFAR-100 | | | |
| | Symmetric | | Flip | |
| | 20% | 50% | 20% | 45% |
| GCE | 63.22±0.45 | 53.16±0.72 | 64.15±0.44 | 40.58±0.49 |
| Forward | 54.90±0.74 | 41.85±0.71 | 56.12±0.54 | 36.88±2.32 |
| Co-teaching | 48.48±0.66 | 36.77±0.52 | 42.79±0.79 | 27.97±0.20 |
| DMI | 62.65±0.39 | 52.42±0.64 | 59.56±0.73 | 38.17±2.02 |
| T-Revision | 62.72±0.69 | 49.12±0.22 | 64.33±0.49 | 41.55±0.95 |
| Dual T | 62.16±0.58 | 52.49±0.37 | 67.21±0.43 | 47.60±0.43 |
| TVD | 63.52±0.40 | 52.54±1.33 | 67.65±0.89 | 56.53±0.16 |
| VolMinNet | 64.94±0.40 | 53.89±1.26 | 68.45±0.69 | 58.90±0.89 |
| ROBOT | 73.03±0.12 | 65.11±0.53 | 75.79±0.51 | 70.20±0.33 |
| SOP | 74.42±0.42 | 66.46±0.65 | 73.93±0.55 | 63.32±0.87 |
| TMR | **75.90±0.23** | **70.93±0.44** | **76.03±0.20** | **70.86±0.56** |

## 4 EXPERIMENTS

In this section, we showcase experimental findings to demonstrate the ability of our proposed method comparing with other methods on diverse noisy datasets.

### 4.1 BENCHMARK DATASETS

We evaluate our method using seven image classification datasets: CIFAR-10 and CIFAR-100 (Krizhevsky et al., 2009), CIFAR-10N and CIFAR-100N (Wei et al., 2021), Clothing1M (Xiao et al., 2015), Webvision and ILSVRC2012 (Li et al., 2017). In addition to common symmetric noise and pair flipping noise, we also incorporate various other types of noise. Specifically, these noise types included: (a) symmetric noise at 20% and 50%; (b) label pair-flipping noise at 20% and 45%; (c) instance-dependent noise. We starte with a 10% noise level and increment it by 10% until reaching

50%. To simulate instance-dependent noise, we employ the noisy data construction same as Xia et al. (2020b). Besides, we utilize real noisy datasets to conduct verification experiments as well. We use ResNet-18 as the backbone network on CIFAR-10 and CIFAR-100, a ResNet-50 pre-trained on Clothing1M and InceptionResNetV2 on Webvision. For more experimental details, including hyperparameter settings, please see Appendix C.

## 4.2 METHOD

We conduct a comparison among the following methods: (1) GCE (Zhang & Sabuncu, 2018), (2) Forward (Patrini et al., 2017), (3) Co-teaching (Han et al., 2018), train two networks. (4) DMI (Xu et al., 2019), (5) T-Revision (Xia et al., 2019), (6) Dual-T (Yao et al., 2020), (7) TVD (Zhang et al., 2021b), (8) VolMinNet (Li et al., 2021b), (9) ROBOT (Yong et al., 2022), (10) SOP (Liu et al., 2022). In addition, transition matrix methods specially designed for instance-dependent label noise: (11) TMDNN (Yang et al., 2022), (12) PartT (Xia et al., 2020b), (13) MEIDTM (Cheng et al., 2022a), and (14) ELR (Liu et al., 2020) on real noisy data. Since our method makes no use of additional techniques, like data augmentation or semi-supervised training (Li et al., 2020; 2023), we do not compare methods using these tricks in our experiments.

Table 2: Test accuracy with instance-dependent noise on CIFAR-10/100.

|  | CIFAR-10 | | | | |
|  | IDN-10% | IDN-20% | IDN-30% | IDN-40% | IDN-50% |
| --- | --- | --- | --- | --- | --- |
| GCE | 90.82±0.05 | 88.89±0.08 | 82.90±0.51 | 74.18±3.10 | 58.93±2.67 |
| Forward | 91.71±0.08 | 89.62±0.14 | 86.93±0.15 | 80.29±0.27 | 65.91±1.22 |
| Co-teaching | 90.80±0.05 | 88.43±0.08 | 86.40±0.41 | 80.85±0.97 | 62.63±1.51 |
| DMI | 91.43±0.18 | 89.99±0.15 | 86.87±0.34 | 80.74±0.44 | 63.92±3.92 |
| VolMinNet | 89.97±0.57 | 87.01±0.64 | 83.80±0.67 | 79.52±0.83 | 61.90±1.06 |
| TMDNN | 90.45±0.72 | 88.14±0.66 | 84.55±0.48 | 79.71±0.95 | 63.33±2.75 |
| PartT | 90.32±0.15 | 89.33±0.70 | 85.33±1.86 | 80.59±0.41 | 64.58±2.86 |
| MEIDTM | 92.91±0.07 | 92.26±0.25 | 90.73±0.34 | 85.94±0.92 | 73.77±0.82 |
| SOP | 93.58±0.31 | 93.07±0.45 | 92.42±0.43 | 89.83±0.77 | 82.52±0.97 |
| TMR | **94.60±0.20** | **93.89±0.17** | **93.04±0.19** | **91.76±0.26** | **88.60±0.46** |
|  | CIFAR-100 | | | | |
|  | IDN-10% | IDN-20% | IDN-30% | IDN-40% | IDN-50% |
| GCE | 69.18±0.14 | 68.35±0.33 | 66.35±0.13 | 62.09±0.09 | 56.68±0.75 |
| Forward | 67.81±0.48 | 67.23±0.29 | 65.42±0.63 | 62.18±0.26 | 58.61±0.44 |
| Co-teaching | 67.91±0.34 | 67.40±0.44 | 64.13±0.43 | 59.98±0.28 | 57.48±0.74 |
| DMI | 67.06±0.46 | 64.72±0.64 | 62.80±1.46 | 60.24±0.63 | 56.52±1.18 |
| VolMinNet | 67.78±0.62 | 66.13±0.47 | 61.08±0.90 | 57.35±0.83 | 52.60±1.31 |
| TMDNN | 68.42±0.42 | 66.62±0.85 | 64.72±0.64 | 59.38±0.65 | 55.68±1.43 |
| PartT | 67.33±0.33 | 65.33±0.59 | 64.56±1.55 | 59.73±0.76 | 56.80±1.32 |
| MEIDTM | 69.88±0.45 | 69.16±0.16 | 66.76±0.30 | 63.46±0.48 | 59.18±0.16 |
| SOP | 74.09±0.52 | 73.13±0.46 | 72.14±0.46 | 68.98±0.58 | 64.85±0.86 |
| TMR | **76.54±0.32** | **75.92±0.44** | **74.74±0.65** | **72.09±0.70** | **69.39±0.66** |

## 4.3 EXPERIMENTAL RESULT

We perform 5 independent runs for each experimental configuration, then present the mean and standard deviation values for methods in Tables 1, 2 and 3. Tables 4 and 5 show the results on three additional real noise datasets. The top-performing methods are indicated in bold in each table.

Table 1 showcases the performance of various contrastive methods on the CIFAR-10 and CIFAR-100 with synthetic symmetric noise and pair-flipping noise. It is evident that the TMR method outperforms all other transition matrix methods significantly on these class-dependent noisy scenarios. SOP (Liu et al., 2022), as a method that also apply implicit regularization approach based on sparsity assumptions, does not perform as well as our method especially when the noise rate is relatively high. Conversely, the proposed TMR method employs the transition matrix to effectively estimate the overall trend and expand ways to use sparsity, thus exhibiting robustness in the presence

of higher noise proportions. For example, on the CIFAR-10 and CIFAR-100 datasets with 45% pair flipping noise, TMR surpasses SOP by 3.03 and 7.54 percentage points, respectively. This clearly demonstrates the effectiveness of our method in handling various class-dependent noise scenarios, even at higher proportions.

Table 2 presents the performance of comparative methods on CIFAR-10 and CIFAR-100 datasets under instance-dependent noise pollution. Many methods based on class-dependent transition matrix fail to provide effective results on instance-dependent noise, like VolMinNet. It can be observed that as the noise ratio increases, the test accuracy of the privous transition matrix methods significantly decreases, even for those specifically designed for instance-dependent noise. Particularly, on CIFAR-100 with 50% IDN data, the test accuracy of all transition matrix methods is below 60%, while our proposed TMR achieves a test accuracy of 69.39, demonstrating excellent performance. Moreover, under high levels of noise, the decline in performance is more pronounced for SOP same as class-dependent noise, whereas TMR maintains robust performance.

Table 3: Test accuracy on CIFAR-10N and CIFAR-100N.

|  | CIFAR-10N | | | | | CIFAR-100N |
|  | Aggregate | Random 1 | Random 2 | Random 3 | Worst | Noisy |
|---|---|---|---|---|---|---|
| CE | 87.77±0.38 | 85.02±0.65 | 86.46±1.79 | 85.16±0.61 | 77.69±1.55 | 50.50±0.66 |
| GCE | 87.85±0.70 | 87.61±0.28 | 87.70±0.56 | 87.58±0.29 | 80.66±0.53 | 56.73±0.30 |
| Forward | 88.24±0.22 | 86.88±0.50 | 86.14±0.21 | 87.04±0.35 | 79.49±0.46 | 57.01±1.03 |
| T-Revision | 88.52±0.17 | 88.33±0.32 | 87.71±1.02 | 87.79±0.67 | 80.48±1.20 | 51.55±0.31 |
| TVD | 88.96±0.15 | 88.41±0.13 | 88.35±0.11 | 88.01±0.40 | 80.15±0.25 | 56.89±0.32 |
| VolMinNet | 89.70±0.12 | 88.30±0.12 | 88.27±0.09 | 88.19±0.41 | 80.53±0.20 | 57.80±0.31 |
| ROBOT | 91.35±0.03 | 90.46±0.18 | 90.37±0.15 | 90.31±0.21 | 84.05±0.33 | 61.25±0.26 |
| SOP | 92.36±0.15 | 91.49±0.17 | 91.88±0.11 | 91.93±0.20 | 86.51±0.30 | 62.73±0.33 |
| **TMR** | **93.60±0.13** | **92.96±0.18** | **93.01±0.09** | **92.74±0.11** | **87.38±0.18** | **64.28±0.27** |

Table 4: Test accuracy on Clothing1M.

| CE | Forward | Co-teaching | DMI | T-Revision | Dual T |
|---|---|---|---|---|---|
| 69.1 | 69.8 | 69.2 | 70.9 | 70.9 | 71.5 |
| TVD | VolminNet | ROBOT | ELR | SOP | TMR |
| 71.7 | 72.4 | 72.7 | 72.9 | 73.5 | **74.2** |

Table 5: Test accuracy on WebVision and ILSVRC2012.

|  | Forward | Co-teaching | VolminNet | ELR | SOP | TMR |
|---|---|---|---|---|---|---|
| WebVision | 61.1 | 63.6 | 68.3 | 76.2 | 76.6 | **77.5** |
| ILSVRC2012 | 57.3 | 61.5 | 64.2 | 68.7 | 69.1 | **71.6** |

Tables 3, 4 and 5 display the results obtained on real-world data. Our approach consistently achieves the best results for real noise, regardless of the type of noisy labels: aggregate, random, or worst for CIFAR-10N, as well as for noisy labels in CIFAR-100N, which includes a larger number of classes. When encountering large datasets like Clothing1M and complex image datasets like Webvision, TMR can also achieve impressive results without additional auxiliary tricks or complex models required. Through extensive experiments conducted on five real datasets, we empirically confirm that TMR is suitable for diverse types of noise.

## 5 CONCLUSION

In summary, we combine the transition matrix and implicit regularization to propose the TMR method. The TMR method overcomes the limitations of the transition matrix in handling instance-dependent noise and improves upon the shortcomings of ordinary implicit regularization in dealing with high-proportion noise. This paper provides a thorough theoretical analysis and finds that the TMR method satisfies statistical consistency and generalization, making it applicable to various types of noise.

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

## A  RELATED WORKS

### A.1  TRANSITION MATRIX METHODS

Most previous transition matrix methods focus on class-dependent label noise to simplify the estimation difficulty. Some of the early methods (Patrini et al., 2017; Xia et al., 2019; Yao et al., 2020) usually assume the existence of anchor points and make the transition matrix identifiable by finding anchor points or approximate anchor points. To mitigate the anchor point assumption, VolMinNet (Li et al., 2021b) and TVD (Zhang et al., 2021b) add different forms of regularization for the transition matrix respectively to make it identifiable. While other methods (Cheng et al., 2022b; Kye et al., 2022) try setting up unique network structure to estimate the transition matrix. Besides, Shu et al. (2020); Yong et al. (2022) utilize structures like meta-learning to estimate the transition matrix, but may require more clean data and computational consumption. Although the above methods are designed to handle class-dependent label noise, it is not suitable when encountering instance-dependent noise or real-world noisy data.

However, it is not feasible to estimate a transition matrix individually for each sample without other assumptions or multiple noisy labels (Liu et al., 2023). In order to achieve an approximate estimation of the instance-dependent transition matrix, Goldberger & Ben-Reuven (2016) uses an adaptation layer to estimate the transition matrix based on each sample's output, but the error is large due to the influence of the initial value. While Yang et al. (2022) uses a separate network to estimate the transition matrix based on the Bayesian label. Some methods (Xia et al., 2020b; Wang et al., 2021; Zhu et al., 2021; 2022) learn a part-dependent or group-dependent matrix through clustering, which is a compromise estimation method lies between instance-dependent and class-dependent methods. Other methods (Cheng et al., 2022a; Jiang et al., 2021) utilize similarity in feature space to assist transition matrix learning. Although these instance-dependent transition matrix methods achieve identifiability through special treatments, they are usually relatively complex and have larger errors, which is contrary to the convenient and simple characteristics of transition matrix methods.

### A.2  IMPLICIT REGULARIZATION

Implicit regularization can be regarded as a statistical method for sparsity, playing the role of minimizing $L_1$ loss in sparse noise learning and being currently used in various models (Zhao et al., 2019; Vaskevicius et al., 2019; You et al., 2020; Li et al., 2021a; Zhao et al., 2022). Among these methods, SOP (Liu et al., 2022) is the one worthy of special attention, which is related to our method. SOP also uses implicit regularization for noisy label learning, which gives a sparse representation of the residual term between prediction and observed noisy label. However, it does not take advantage of the overall transfer probability of noise and the noise sparsity assumption does not apply to high noise rates situation, so its performance on large noise rates data is relatively weak. We will compare it with our proposed method by experimental results specifically in Section 4.

# B ALGORITHM AND PROOFS

## B.1 ALGORITHM

The steps of our TMR algorithm are shown in detail in Algorithm 1

---

**Algorithm 1** Transition Matrix with Implicit Regularization (TMR)

---

**Input:** Training data $\{(\boldsymbol{x}_i, \boldsymbol{y}_i)\}_{i=1}^N$, network $f_{\boldsymbol{\theta}}(\cdot)$, coefficient $\lambda$, learning rate $\tau_{\boldsymbol{\theta}}, \tau_{\boldsymbol{u}}, \tau_{\boldsymbol{v}}, \tau_{\boldsymbol{T}}$, batch size $m$, epoch number $E$, transition matrix update frequency $k$.

**Initialization:** Transition matrix $\boldsymbol{T}$ with an identity matrix, draw entries of $\{\boldsymbol{u}_i, \boldsymbol{v}_i\}_{i=1}^N$ from i.i.d. Gaussian distribution with zero-mean and s.t.d. 1e-8.

**for** $t = 1$ **to** $E$ **do**
   **for** $b = 1$ **to** $N/m$ **do**
      Get a sample batch $\mathcal{B} \subseteq \{1, \ldots, N\}$ with $|\mathcal{B}| = m$
      Calculate loss $\mathcal{L}$ by 7 with batch $\mathcal{B}$
      **for** $i$ in $\mathcal{B}$ **do**
         Update $\boldsymbol{u}_i \leftarrow \boldsymbol{u}_i - \tau_{\boldsymbol{u}} \cdot \partial \mathcal{L} / \partial \boldsymbol{u}_i$
         Update $\boldsymbol{v}_i \leftarrow \boldsymbol{v}_i - \tau_{\boldsymbol{v}} \cdot \partial \mathcal{L} / \partial \boldsymbol{v}_i$
      **end for**
      Update $\boldsymbol{\theta} \leftarrow \boldsymbol{\theta} - \tau_{\boldsymbol{\theta}} \cdot \partial \mathcal{L} / \partial \boldsymbol{\theta}$
      **if** $b/k$ is 0 **then**
         Update $\boldsymbol{T} \leftarrow \boldsymbol{T} - \tau_{\boldsymbol{T}} \cdot \partial \mathcal{L} / \partial \boldsymbol{T}$
      **end if**
   **end for**
**end for**

**Output:** Network parameters $\hat{\boldsymbol{\theta}}$, variables $\{\hat{\boldsymbol{u}}_i, \hat{\boldsymbol{v}}_i\}_{i=1}^N$ and transition matrix $\hat{\boldsymbol{T}}$.

---

## B.2 CONDITIONS

**Condition 1.** *For optimization problem 20, initialize parameters in the algorithm 1 with $\boldsymbol{\theta} = \boldsymbol{0}$, $\boldsymbol{u}_i = t\boldsymbol{1}$, $\boldsymbol{v} = t\boldsymbol{1}$, where $\boldsymbol{0}, \boldsymbol{1}$ are vectors of all 0 or 1 respectively, $t$ is a small value scalar. There exists a given $\alpha_0 > 0$ such that the learning rates of gradient descent satisfy $lr(\boldsymbol{u}) = lr(\boldsymbol{v}) = \alpha lr(\boldsymbol{\theta})$, $\alpha < \alpha_0$.*

**Condition 2.** *Denote the rank of $\boldsymbol{G}$ in 18 as $r$, the number of sparse nonzero entries of $\boldsymbol{R}_*$ is $k$, $\boldsymbol{P}$ is the matrix of row vectors in SVD decomposition of $\boldsymbol{G}$. Define $s = \frac{N}{r} max_{1 \leq i \leq N} \|\boldsymbol{P}^\top \boldsymbol{e}_i\|_2^2$. Then $k, r, s$ satisfy $4k^2 rs < N$.*

**Condition 3.** *The row vectors of matrix $\boldsymbol{F}$ in 17 are sufficiently scattered, which is a weakened requirement of the anchor points assumption can be found in Definition 2 of Li et al. (2021b).*

## B.3 PROOF OF THEOREM 1

*Proof.* Denote $\boldsymbol{Q} = (\boldsymbol{I}_C \otimes \boldsymbol{\theta}) \cdot \boldsymbol{T}$, the optimization problem in 20 can be written as:

$$\min \frac{1}{2} \|\boldsymbol{G} \cdot \boldsymbol{Q} + \boldsymbol{U} \odot \boldsymbol{U} - \boldsymbol{V} \odot \boldsymbol{V} - \tilde{\boldsymbol{Y}}\|_2^2 + \lambda \cdot \log \det(\boldsymbol{T}). \tag{23}$$

Since implicit regularization can minimize the $L_1$ loss and according to Proposition 3.3 in Liu et al. (2022), the first half of 23 will converge to a global solution for any fixed $\boldsymbol{T}$ under Condition 1. Furthermore, it can be converted into the following optimization problem:

$$\min_{\boldsymbol{Q}, \boldsymbol{R}} \frac{1}{2} \|\boldsymbol{Q}\|_2^2 + \beta \|\boldsymbol{R}\|_1, \quad \text{s.t.} \quad \tilde{\boldsymbol{Y}} = \boldsymbol{G} \cdot \boldsymbol{Q} + \boldsymbol{R}, \tag{24}$$

where $\beta = -\frac{\log t}{2\alpha}$ as defined in 1. When Condition 2 is true, the solution to 24 are $\boldsymbol{Q}_*$ and $\boldsymbol{R}_*$, where $\tilde{\boldsymbol{Y}}$ is produced by $\boldsymbol{G} \cdot \boldsymbol{Q}_* + \boldsymbol{R}_*$. This conclusion can be deduced from the analogy of Proposition 3.5 in Liu et al. (2022). Combining 18, we can get:

$$\boldsymbol{Q}_* = (\boldsymbol{I}_C \otimes \boldsymbol{\theta}_*) \cdot \boldsymbol{T}_*. \tag{25}$$

Therefore, problem 23 transform into an optimization problem with parameter $\boldsymbol{\theta}, \boldsymbol{T}$:

$$\min_{\boldsymbol{\theta}, \boldsymbol{T}} \log \det(\boldsymbol{T}), \quad \text{s.t.} \quad (\boldsymbol{I}_C \otimes \boldsymbol{\theta}) \cdot \boldsymbol{T} = \boldsymbol{Q}_*. \tag{26}$$

The above optimization problem has the same form as optimization problem 3. It can be seen from the Theorem 1 in Li et al. (2021b), under Condition 3, the solution to problem 26 is:

$$\hat{\boldsymbol{\theta}} = \boldsymbol{\theta}_*, \quad \hat{\boldsymbol{T}} = \boldsymbol{T}_*. \tag{27}$$

To sum up, when all conditions in Appendix B.2 are met, we can get the ground truth solution $\boldsymbol{\theta}_*$, the estimators by our algorithm converge to $\boldsymbol{T}_*, \boldsymbol{R}_*$ as mentioned in Theorem 1. □

### B.4 PROOF OF THEOREM 2

*Proof.* We use the inequality we use Hoeffding inequality (Hoeffding, 1994) to help us complete the proof. Since $\hat{\boldsymbol{\theta}}_{(N)}, \hat{\boldsymbol{T}}_{(N)}$ are not independent of the samples, we use $\epsilon$-cover as mentioned in Section 3.2 to deal with the problem. In addition, the parameter $\boldsymbol{R}$ is omitted in the following proof for convenience and does not affect the understanding of the results.

According to the definition of $\epsilon$ covering, We can find a pair of parameters $\boldsymbol{\theta}_k, \boldsymbol{T}_k$ in the covering set such that:

$$|\ell(\boldsymbol{\theta}_k, \boldsymbol{T}_k; X, Y) - \ell(\hat{\boldsymbol{\theta}}_{(N)}, \hat{\boldsymbol{T}}_{(N)}; X, Y)| \leq \epsilon, \forall (X, Y) \in \mathcal{X} \times \mathcal{Y}. \tag{28}$$

Average the loss over samples, we have:

$$\mathcal{L}(\hat{\boldsymbol{\theta}}_{(N)}, \hat{\boldsymbol{T}}_{(N)}; \tilde{\mathbb{D}}) \leq \mathcal{L}(\boldsymbol{\theta}_k, \boldsymbol{T}_k; \tilde{\mathbb{D}}) + \epsilon. \tag{29}$$

To meet the requirement of probability $1 - \delta$ in Theorem 2, we take the probability value as $\delta/2\mathcal{N}_\mathcal{F}$ in Hoeffding inequality due to the randomness of $k$. Thus, with probability at least $1 - \delta/2\mathcal{N}_\mathcal{F}$,

$$\mathcal{L}(\boldsymbol{\theta}_k, \boldsymbol{T}_k; \tilde{\mathbb{D}}) \leq \mathcal{L}(\boldsymbol{\theta}_k, \boldsymbol{T}_k; \tilde{\mathbb{D}}_{(N)}) + M\sqrt{\frac{\ln(2\mathcal{N}_\mathcal{F}/\delta)}{2n}}. \tag{30}$$

By the definition of 28,

$$\mathcal{L}(\boldsymbol{\theta}_k, \boldsymbol{T}_k; \tilde{\mathbb{D}}_{(N)}) \leq \mathcal{L}(\hat{\boldsymbol{\theta}}_{(N)}, \hat{\boldsymbol{T}}_{(N)}; \tilde{\mathbb{D}}_{(N)}) + \epsilon. \tag{31}$$

According to the property of $\hat{\boldsymbol{\theta}}_{(N)}, \hat{\boldsymbol{T}}_{(N)}$ in 21, for any $\boldsymbol{\theta} \in \mathbb{R}^p, \boldsymbol{T} \in \mathbb{T}$,

$$\mathcal{L}(\hat{\boldsymbol{\theta}}_{(N)}, \hat{\boldsymbol{T}}_{(N)}; \tilde{\mathbb{D}}_{(N)}) \leq \mathcal{L}(\boldsymbol{\theta}, \boldsymbol{T}; \tilde{\mathbb{D}}_{(N)}) + \epsilon. \tag{32}$$

Using the Hoeffding inequality again with probability $\delta/2$, with probability at least $1 - \delta/2$ we have:

$$\mathcal{L}(\boldsymbol{\theta}, \boldsymbol{T}; \tilde{\mathbb{D}}_{(N)}) \leq \mathcal{L}(\boldsymbol{\theta}, \boldsymbol{T}; \tilde{\mathbb{D}}) + M\sqrt{\frac{\ln(2/\delta)}{2n}}. \tag{33}$$

Combining inequalities 29, 30, 31, 32, 33 and adding the probability values, we get the conclusion that with probability at least $1 - \delta$,

$$\mathcal{L}(\hat{\boldsymbol{\theta}}_{(N)}, \hat{\boldsymbol{T}}_{(N)}; \tilde{\mathbb{D}}) \leq \mathcal{L}(\boldsymbol{\theta}, \boldsymbol{T}, ; \tilde{\mathbb{D}}) + M\sqrt{\frac{\ln(2\mathcal{N}_\mathcal{F}/\delta)}{2n}} + M\sqrt{\frac{\ln(2/\delta)}{2n}} + 3\epsilon, \forall \boldsymbol{\theta} \in \mathbb{R}^p, \boldsymbol{T} \in \mathbb{T}. \tag{34}$$

□

## C EXPERIMENT DETAILS

### C.1 EXPERIMENTAL SETUP

We conduct experiments on a single NVIDIA 3090Ti graphics card. For software, we use Python 3.11 and PyTorch 1.10 to build the models. Throughout the training process, transition matrix updates are carried out using the Adam optimization method, while updates for other parameters are performed using the stochastic gradient descent (SGD) optimization method. The experimental setup involves a few training hyper-parameters, including the backbone network used, batch size, learning rate for parameters, and weight of the regularization term. For specific experimental configurations, please refer to Table 6 in Appendix C.2.

## C.2 HYPER-PARAMETERS SETTING

The backbone network and hyper-parameters of the experiments on each dataset are listed in the table 6.

Table 6: Hyper-parameters on CIFAR-10/100, Clothing-1M and Webvision.

| | CIFAR-10 | CIFAR-100 | Clothing1M | Webvision |
|---|---|---|---|---|
| Network | ResNet18 | ResNet18 | ResNet-50 | InceptionResNetV2 |
| Batch size | 128 | 128 | 64 | 32 |
| Training samples | 50,000 | 50,000 | 1,000,000 | 65,944 |
| Epochs | 300 | 300 | 10 | 100 |
| Learning rate(lr) for network | 0.05 | 0.05 | 0.002 | 0.02 |
| lr decay for network | Cosine | Cosine | 5th | 50th |
| Weight decay for network | 5e-4 | 5e-4 | 1e-3 | 5e-4 |
| lr for $T$ | 0.0005 | 0.0002 | 0.0001 | 0.0005 |
| lr decay for $T$ | 30th, 60th | 30th, 60th | 5th | 50th |
| Initialization for $T$ | -2 | -4.5 | -2.5 | -4 |
| lr for $u, v$ | 10, 10 | 1, 100 | 0.1, 1 | 0.1, 1 |
| lr decay for $u, v$ | Cosine | Cosine | 5th | 50th |
| Coefficient $\lambda$ | 0.001 | 0.001 | 0.001 | 0.001 |

