# OpenReview forum: "Incorporating Implicit Regularization to Enhance the Transition Matrix Method for Effective Handling of Diverse Label Noise"
_ICLR.cc/2024/Conference — Submitted to ICLR 2024_

### Official Review · Reviewer_q63a · 2023-10-13

**Soundness:** 2 fair
**Presentation:** 3 good
**Contribution:** 2 fair
**Rating:** 5
**Confidence:** 5

**Summary:**

This paper focuses on label-noise learning, which is a realistic and important topic in weakly supervised learning. Specifically, this paper mainly targets the estimation of the noise transition matrix. It argues that estimating the transition matrix for each instance may be unidentifiable and computationally expensive in the case of instance-dependent label noise and real-world situations. Therefore, it proposes to learn the residual between probabilities, which is easier. Theoretical analysis is provided to discuss the estimation performance and generalization performance. Experiments on both synthetic and real-world label-noise datasets demonstrate the effectiveness of the proposed method.

**Strengths:**

- The motivation is clear. It is significant to study how to estimate the noise transition matrix, especially for the instance-dependent transition matrix.
- Experimental results are overall great. On a series of tasks, the proposed method achieves the best performance.

**Weaknesses:**

- The contributions are somewhat overclaimed.
- The theoretical analysis should be improved. More descriptions and explanations should be provided.
- The writing also should be polished. For the current form, there are a series of unclear justifications.

More details about the above weaknesses can be checked below.

**Questions:**

**On contributions**
- This paper claims that it contributes to estimating instance-dependent transition matrix. However, after reading this paper, it seems that this paper studies a specific type of label noise, which is a weak version of instance-dependent label noise. Specifically, the paper proposes first to learn a global transition matrix and learn the residual term with respect to each instance. This holds only when the noise patterns of different instances are similar.
- The previous work [R1] employs a similar idea. However, [R1] is not discussed.

**On theoretical analysis**
- The analysis highly relies on neural tangent kernels. It is somewhat less general for me.
- For Theorem 2, when the sample size $n$ approximates $+\infty$, the error is not zero. It depends on the parameter $\epsilon$.
- Condition 1 is not very reasonable. It needs the network parameter to be zero at the beginning of training. It is not very practical.


**On writing**
- In the “Introduction”, the definitions of $Y$ and $X$ should be provided.
- This paper argues that compared to traditional transition matrix methods for class-dependent label noise, the proposed method does not require much additional time consumption. However, I do not find strong evidence about this claim.
- The notation $-\top$ is a bit confusing. "Inverse + Transpose"?
- Why using $u$ and $v$? It is confusing for me.
- Could the proposed method be boosted by semi-supervised learning methods, e.g., DivideMix, for better performance?
----
[R1] Shikun Li et al. Transferring annotator- and instance-dependent transition matrix for learning from crowds. arxiv preprint arXiv:2306.03116.

---

> ### Author Response · Authors · 2023-11-15
>
> Dear reviewer,
>
> Thank you for your review of our paper and providing constructive questions and suggestions. We will address each of the questions you raised and are happy to continue further discussion on related issues with you.
>
> $\textbf{For contributions:}$
>
> The first point:
>
> Specifically, our method has theoretical analysis results for the specific noise patterns you mentioned. However, it is a misunderstanding that our model is not applicable to other instance-dependent noise. The main objective of our paper is to develop a lightweight and practical model that performs well on a broader range of data. Our experimental results indicate that even on more complex instance-dependent data and real-world datasets, our method is more effective compared to other methods of the same type. This demonstrates that despite not fully conforming to our model assumptions, our model is still effective in fitting and learning. (Similar to that you can fit all data with a linear model, whether the effect is appropriate is the key.) Our model exhibits good results for various types of noisy datasets without the need for additional complex structures or techniques, which we believe is meaningful.
>
> The second point:
>
> We have previously read the work you mentioned, which focuses specifically on the transition matrix method in the context of multiple annotators. However, in our considered problem, each sample has only one label, so the research questions and the datasets used in our work are different. Therefore, we did not engage in a comparative discussion with that particular method. The baseline method of this method is TMDNN ([1]), it is an instance-dependent method. We have conducted a detailed comparison between our method and TMDNN in our paper, and our method outperforms it to a great extent.
>
> $\textbf{For theoretical analysis: }$
>
> The first point:
>
> Our method primarily leverages analytical approaches concerning transition matrix and implicit regularization. For a better understanding of the analytical framework and some conclusions, we recommend referring to articles [2] and [3]. These articles provide valuable insights and can enhance comprehension of our analysis approach.
>
> The second point:
>
> The term $\epsilon$ can be arbitrarily small here, as guaranteed by the convergence of Theorem 1. Therefore, the $\epsilon$ term on the right-hand side of the inequality in Theorem 2 can be controlled. If your question is about the tightness of the $\epsilon$-cover control, a more rigorous result can be obtained using Rademacher Complexity [4] or Dudley's inequality [5] from high-dimensional probability theory. This derivation would be relatively complex, and we can provide it in the formal version of the paper.
>
> The third point:
>
> What you said is correct that network parameters are not typically initialized from zero. However, the condition we propose in this context is derived following the simplification of the model analysis, similar to the simplification from Equation (9) to Equation (10). In reality, any initialization is valid, it does not affect our analysis results albeit with a constant term difference. We made this simplification for the sake of ease in derivation and writing, without the requirement limited to this condition in practical application.

---

> ### Author Response · Authors · 2023-11-15
>
> $\textbf{For writing: }$
>
> The first point:
>
> We have provided detailed definitions of X and Y in preliminaries (Section 2.1). However, we acknowledge your suggestion that it would be appropriate to provide them in the initial introduction part as well.
>
> The second point:
>
> Intuitively, our model primarily utilizes the backbone network as main framework without employing time-consuming techniques such as sample selection, contrastive learning for feature training, or graph model construction. To illustrate, we conducted tests on the CIFAR-100 dataset to record the running time for some methods mentioned in our paper with a single NVIDIA 3090Ti, and the results are as follows. We observed that our method does not significantly increase the time consumption compared to those previous transition matrix methods for class-dependent noise. In contrast, approaches that are designed to handle instance-dependent noise or methods with more complex model exhibit higher time consumption.
>
> | CE | VolMinNet | ROBOT | SOP | TMR(Ours) | TMDNN | MEIDTM | ELR+ | DivideMix |
> |--------|--------|--------|--|--|--|--|--|--|
> |1.5h | 1.6h | 6.6h | 1.8h | 1.9h | 3.7h | 5.4h | 4.5h | 8.9h|
>
>
> The third point:
>
> The symbol $-\top$ denotes the transpose of the inverse matrix, i.e., what you understand "Inverse + Transpose". We should provide a specific explanation of this notation in the paper. Thank you for pointing out this issue.
>
> The fourth point:
>
> The form of $u \odot u-v \odot v$ have served as an implicit regularization under specific conditions (see Appendix B.2). It is a technique that has been widely used before, e.g., [6], [7] and [3]. This approach can approximate the effect of constraining the L1 norm to achieve sparsity without the need for an additional regularization term and easier to optimize training than directly use L1 regularization. You can have a certain understanding through these articles and a detailed introduction about it can be seen in Appendix A.2 in our paper.
>
>
> The fifth point:
>
> Our method is a plug-and-play model, which can be combined with any other SOTA technologies, such as semi-supervised, or can be used as a downstream task of contrastive learning, to obtain better results. This does not conflict. However, the core purpose of our paper is to build a relatively simple framework that can achieve good results on various types of noise data. What we pursue is the simplicity of transition matrix methods and the theoretical analysis under certain assumptions. Therefore, we only compare our method with the same type of works, and the combination with other relatively complex SOTA technologies is not the focus of this paper.
>
> Once again, we sincerely appreciate your patient review and suggestions. We hope that the additional explanations above can address your concerns and receive your approval.
>
> [1] Shuo Yang, Erkun Yang, Bo Han, Yang Liu, Min Xu, Gang Niu, and Tongliang Liu. Estimating instance-dependent bayes-label transition matrix using a deep neural network. In International Conference on Machine Learning, pp. 25302–25312. PMLR, 2022.
>
> [2] Xuefeng Li, Tongliang Liu, Bo Han, Gang Niu, and Masashi Sugiyama. Provably end-to-end label noise learning without anchor points. In International conference on machine learning, pp. 6403– 6413. PMLR, 2021b.
>
> [3] Sheng Liu, Zhihui Zhu, Qing Qu, and Chong You. Robust training under label noise by over parameterization. In International Conference on Machine Learning, pp. 14153–14172. PMLR, 2022.
>
> [4] Bartlett, P. L., & Mendelson, S. (2002). Rademacher and Gaussian complexities: Risk bounds and structural results. Journal of Machine Learning Research, 3(Nov), 463-482.
>
> [5] Mendelson, S. (2014). Learning without concentration. Journal of Machine Learning Research, 15(1), 589-613.
>
> [6] Peng Zhao, Yun Yang, and Qiao-Chu He. Implicit regularization via hadamard product over parametrization in high-dimensional linear regression. arXiv preprint arXiv:1903.09367, 2(4): 8, 2019.
>
> [7] Jiangyuan Li, Thanh Nguyen, Chinmay Hegde, and Ka Wai Wong. Implicit sparse regularization: The impact of depth and early stopping. Advances in Neural Information Processing Systems, 34: 28298–28309, 2021a.

---

### Official Review · Reviewer_HJZZ · 2023-10-17

**Soundness:** 2 fair
**Presentation:** 3 good
**Contribution:** 2 fair
**Rating:** 3
**Confidence:** 4

**Summary:**

This paper proposes a new method for estimating a global transition matrix for dealing with noisy labels, which combines implicit regularization to replace the estimation of the individual transition matrix for each example. The proposed approach is claimed to be suitable for diverse types of noise as well as alleviating the problem of inaccurate posterior probability estimation. The authors theoretically analyze the consistency and generalization results of the proposed method, and also conduct experiments on synthetic and real-world datasets with different types of label noise.

**Strengths:**

1.	The paper is written in a clear way.
2.	The experimental results reflect the effectiveness of the proposed algorithm.

**Weaknesses:**

1.	I feel that the high-level idea of this paper is a bit similar to T-revision (Are anchor points really indispensable in label-noise learning? NIPS 19). In T-revision, the authors propose to learn \deltaT which is imposed to transition matrix T. In this paper, the authors aim to learn \gamma(X) imposed on T^{\top}*P(Y|X). Note that these two formations are actually the same, if we let \gamma(X)=\deltaT*P(Y|X). In other words, both methods aim to learn a residual term to correct the original estimation. Therefore, I feel that the authors need to clarify the essential difference between the two methods.
2.	One selling point of this proposed method is that it can handle different types of noise, especially instance-dependent noise, as claimed by the authors. After investigating the model, I think such merits mainly come from the term \gamma(X). However, l cannot fully understand why introducing such term can enable the method to deal with various noise types. The authors claim that they impose sparsity to \gamma(X) to achieve this target, but why? What is the relationship of sparsity with the noise types, especially instance-dependent label noise? I think such rationale is neither clear nor straightforward enough.
3.	Even the formulation (5) is correct, the authors decompose \gamma(X) into N pairs of {u_i, v_i}. Then the problem comes. I notice that no regularization is imposed to {u_i, v_i}, how to guarantee its identifiability, uniqueness, or even optimality? Note that the identifiability of T is very important for label noise learning, and the identifiability of {u_i, v_i} is actually very related to it.
4.	The authors might misused the terms “sample” and “example”, which confused me a lot when I read the paper at the first time. Note that their meanings are totally different. I think most of the term “sample” should be “example” in this paper.
5.	The notion “diagonally dominant” is not correctly used in this paper. Note that when we say a c*c matrix T is diagonally dominant, it means T_ii>|T_i1|+|T_i2|+…|T_i,i-1| +|T_i,i+1| +|T_ic| mathematically, rather than T_ii>T_ij for j \neq i.
6.	The experimental results reveal that sometimes the accuracy of the proposed method is very similar to baseline methods. Therefore, I think statistical significance analysis is needed to justify the real superiority of the proposed method to baseline methods.

**Questions:**

See the weakness part.

---

> ### Author Response · Authors · 2023-11-15
>
> Dear reviewer,
>
> Thank you for your review of our paper and providing constructive questions and suggestions. We will address each of the questions you raised and are happy to continue further discussion on related issues with you.
>
> For Weaknesses:
>
> 1.
> Our method and the T-revision are two completely different approaches. T-revision adjusts the estimation of the transition matrix $\boldsymbol{T}$ by $\boldsymbol{T}+\delta \boldsymbol{T}$, which essentially constructs a transition matrix (independent of feature X) to handle class-dependent noise problems without utilizing the information of feature X to $\boldsymbol{T}$. Therefore, it is not suitable for instance-dependent noise problems. In contrast, our method directly fits the residual term, which contains information from X, rather than directly utilizing information from $ P(\boldsymbol{Y} \mid X)$. Thus, our method is not limited to class-dependent noise problems and can be extended to instance-dependent noise. If, as you mentioned, representing the form $\gamma(X)=(\delta\boldsymbol{T})^{\top} P(\boldsymbol{Y} \mid X)$, where $\delta\boldsymbol{T}$ is independent of X, and $ P(\boldsymbol{Y} \mid X)$ is unknown and need to be estimated by networks. In this case, adding $\gamma(X)$ will not provide extra improvement for learning $\boldsymbol{T}$ and $P(\boldsymbol{Y} \mid X)$ compared to directly using the transition matrix. Experimental results also demonstrate that our method substantially better than T-revision on various datasets, especially in instance-dependent noise scenarios, indicating the fundamental differences between these two methods.
>
> 2.
> We would like to introduce the origin of our ideas here, which may help you understand our method. Our core idea is to construct a lightweight framework based on the transition matrix method, which can transition from solving class-dependent noise to solving instance-dependent noise. Since it is difficult to learn the matrix $\boldsymbol{T}(X)$ (depend on X) for handling instance-dependent noise, we hope to combine the transition matrix and implicit regularization techniques as substitute. We aim to estimate an overall transition matrix similar to the previous class-dependent methods, but it is not enough to handle various complex types of noise. However, on the other hand, after the real label distribution $ P(\boldsymbol{Y} \mid X)$ undergoes an overall transfer by the transition matrix $\boldsymbol{T}$, the difference between $ \boldsymbol{T}^{\top} P(\boldsymbol{Y} \mid X)$ and the noise label distribution  $ P(\tilde{\boldsymbol{Y}} \mid X)$  should not be large. Therefore, we have the idea of through constraining the norm of the residual term to construct the model. Inspired by the idea from previous work using sparsity to deal with noise problems (e.g. [1], [2]), a sparse structure is used to make the residuals relatively small, which leads to our method TMR and we find it is widely effective. In fact, we have also tried to restrict the L2 norm of the residual term, but the results were not as good as now. If the observed noisy label data can be regarded as generated in this way (through a transition matrix and sparse residual term), we can theoretically demonstrate the effectiveness of our method (Theorem 1). In practical applications, experiments have shown that many instance-dependent noise data or real-world datasets can be considered as approximations of our assumption due to our method performs better than these specially designed instance-dependent transition matrix methods. This means that although the assumption of sparse residual may not always hold, the general fit of our framework for various types of noisy label data is valuable.
>
>
> 3.
> The form of $u_i \odot u_i-v_i \odot v_i$ have already served as an implicit regularization under specific conditions (see Appendix B.2). It is a technique that has been widely used before (e.g., [2], [3], [4], detailed introduction about it can be seen in Appendix A.2). This approach can approximate the effect of constraining the L1 norm without the need for an additional regularization term and easier to optimize training than directly use L1 regularization, hence it called "implicit" regularization. In Theorem 1 of our paper, we have demonstrated the identifiability of $u_i$, $v_i$, the transition matrix $\boldsymbol{T}$ and network parameters $\theta$ under specific conditions.
>
> 4.
> In our paper, the term "sample" is used to denote random instance from dataset that lack referential meaning and do not require a specific selection process. Does this align with your understanding of the term "example"? In reality, we have confusion in the interchangeable usage of these two terms in related papers within the field. Could you provide clarification on the distinctions between them? If indeed a misuse exists, we will modify it.

---

> ### Author Response · Authors · 2023-11-15
>
> 5.
> Our definition is followed by the definition provided in [5], which has been more commonly used within label noise learning differs from the mainstream definition in matrix analysis. However, your suggestion is valid, and ambiguous definitions should be eliminated. We will remove this usage from our paper. Thank you for pointing that out.
>
> 6.
> In fact, the experimental results clearly demonstrate the significant improvement of our method compared to the baseline methods across various datasets. We are not quite clear about what you mean by "similar” effects. If you are referring to the comparison with the SOP method (which also utilizes implicit regularization), it is true that the improvement may not be evident at low noise rates. However, our method outperforms SOP noticeably at high noise rates, as we have extensively analyzed and discussed in our paper.
>
> Once again, we sincerely appreciate your patient review and suggestions. We hope that the additional explanations above can address your concerns and receive your approval.
>
> References
>
> [1] Zhou, Xiong, et al. "Learning with noisy labels via sparse regularization." Proceedings of the IEEE/CVF international conference on computer vision. 2021.
>
> [2] Liu, Sheng, et al. "Robust training under label noise by over-parameterization." International Conference on Machine Learning. PMLR, 2022.
>
> [3] Peng Zhao, Yun Yang, and Qiao-Chu He. Implicit regularization via hadamard product over parametrization in high-dimensional linear regression. arXiv preprint arXiv:1903.09367, 2(4): 8, 2019.
>
> [4] Jiangyuan Li, Thanh Nguyen, Chinmay Hegde, and Ka Wai Wong. Implicit sparse regularization: The impact of depth and early stopping. Advances in Neural Information Processing Systems, 34: 28298–28309, 2021a.
>
> [5] Xuefeng Li, Tongliang Liu, Bo Han, Gang Niu, and Masashi Sugiyama. Provably end-to-end label noise learning without anchor points. In International conference on machine learning, pp. 6403– 6413. PMLR, 2021b.

---

### Official Review · Reviewer_11Gr · 2023-10-30

**Soundness:** 2 fair
**Presentation:** 2 fair
**Contribution:** 3 good
**Rating:** 5
**Confidence:** 5

**Summary:**

This paper proposed integrating implicit regularization and the transition matrix method for noisy label learning. To avoid the difficulty of estimating the instance-dependent transition matrix, by assuming the sparsity of the difference between the estimated posterior probability distribution and the noisy label distribution, it only estimated a global transition matrix for each sample, and an implicit regularization is applied in the residual vector to promote the sparsity.  Theoretical analyses provided the consistency and generalization results of the proposed method. Experimental results confirm the superiority of the proposed method.

**Strengths:**

1. The combination of transition matrix methods and other robust techniques is a seldom-explored and important direction for noisy label learning.
2. This work provided the consistency and generalization results of the proposed method, which keeps the advantages of the transition matrix methods.
3.  Experimental results showed promising results on various benchmarks.

**Weaknesses:**

1. The specific definition of the sparsity of the residual term in this paper is unclear.  Does it mean that the residual term includes many zero elements?  Besides, could the authors provide some evidence to support the sparsity assumption across various noisy cases? I think it's necessary to show the advantages of the assumptions of the proposed method compared with existing methods.
2. Some statements in this paper need further discussion or clarification:
- Handling diverse label noise doesn't make sense in my opinion, since the real-world label noise is usually instance-dependent.
- What is a "valid" transition matrix and residual term in Section 2.2? Could the authors provide some theoretical results that show in certain cases, a clean class-posterior probability can be obtained, regardless of instance-dependent noise or the noisy class-posterior has a large estimation error?
- Why log det(T) can be ignored in the generalization analysis？
3. (Minor) I suggest the authors discuss more recent SOTA works, e.g. [3,4].
4. (Minor) The novelty of techniques seems a little limited. It seems that the proposed method mainly combined the techniques from [1] and [2].
5. (Minor) The presentation should be improved largely. For example, this paper only uses a single number to refer to one equation.

[1] Provably end-to-end label-noise learning without anchor points. ICML 2021

[2] Robust training under label noise by over-parameterization. ICML 2022

[3] Selective-supervised contrastive learning with noisy labels. CVPR 2022

[4] Instance-dependent noisy label learning via graphical modelling. WACV 2023

**Questions:**

See above weaknesses. I am happy to increase my score if my concerns are addressed.

---

> ### Author Response · Authors · 2023-11-15
>
> Dear reviewer,
>
> Thank you for your review of our paper and providing constructive questions and suggestions. We will address each of the questions you raised and are happy to continue further discussion on related issues with you.
>
> For Weaknesses:
>
> 1.
> The residual term refers to the difference between the true posterior probability of the noise data $ P(\tilde{\boldsymbol{Y}} \mid X)$ and the probability distribution after multiplication by the transition matrix $ \boldsymbol{T}^{\top} P(\boldsymbol{Y} \mid X)$. In fact, this residual term is not necessarily sparse, but we use a sparse structure to fit this residual term in the paper. The idea to use sparse residual term comes from previous work using sparsity to deal with noise problems (e.g. [1], [2]). We hope to fit the noisy label through a suitable transition matrix for the overall data combined with this sparse term. Theorem 1 is to illustrate that when the source of noise data can be regarded as or approximated as being generated in this way, our method can theoretically achieve optimal learning effects. In the experimental part, the experimental results on various types of noisy label data show that our method is universal, and it outperforms other existing transition matrix methods, regardless of class-dependent label noise, instance-dependent label noise, or unknown real-world datasets. (This does not mean that our model assumptions are always correct, but compared to other methods, it has a better effect of fitting noisy label data.)
>
> 2.
>
> The first point:
>
> As mentioned above, we aim use experiments to show that our proposed method is not only effective under the assumption of our model, but also universally applicable to various types of noisy data and exhibits high practical reliability. Moreover, the experimental results indicate that compared to other transition matrix methods, we can outperform those specifically designed methods for specific type of label noise on corresponding datasets.
>
> The second point:
>
> The “valid” transition matrix and sparse residual term refer to our desire to fit the noisy label distribution $ P(\tilde{\boldsymbol{Y}} \mid X)$ using a model that multiplies a transition matrix $\boldsymbol{T}$ with the clean label distribution $ P(\boldsymbol{Y} \mid X)$ (learned through the network) and adds a sparse term $\gamma(X)$. If the observed noisy label data is indeed generated or can be approximately viewed in this way, we have conducted analysis in the theoretical section. Theorem 1 provides the conclusion that our method can learn network parameters that are statistically consistent with the parameters learned with clean label data, which means the ability to learn clean label distributions through network. In other unknown situations, there is no direct theoretical analysis results for our model, but this does not mean that our method is limited. Experimental results show that our method performs better than existing transition matrix methods on either synthetic instance-dependent noisy data or real-world noisy datasets. Therefore, our model is effective in practice. (That means our model may not be accurate for all noisy label data, but it is universally effective and concise.)
>
> The third point:
>
> This is because this part $\log det(\boldsymbol{T})$ does not vary with the sample X, so its impact on both sides of the inequality for equation (21) and equation (22) are same. In fact, this assumption in the text is made for simplifying the loss function in the formula derivation and adding it up will not affect the result.
>
> 3.
> Thank you for your suggestion. In reality, we have already read those papers as you mentioned before writing our paper. However, they all use more complex model structures such as contrastive learning and graph models. The original idea of our paper is to propose a lightweight unified model framework that is more widely applicable to various types of label noise, based on the theoretical analysis of transition matrix methods. Therefore, we did not compare our work with those SOTA works because we are on different tracks. We specifically compared it within the works based on transition matrix. It is also worth mentioning that our method can be well integrated with those SOTA techniques, such as downstream tasks in contrastive learning, but this is not the focus of our paper.

---

> ### Author Response · Authors · 2023-11-15
>
> 4.
> If only considering the form of the loss function, our method does look like a combination of VolMinNet[3] and SOP[2]. However, it is not simply a patchwork. Our core idea is to propose a relatively concise unified approach framework that can perform well on a wider range of noisy datasets. The analytical ideas are presented in the methods section. We are the first to propose the idea of combination transition matrix and implicit regularization to deal with the label noise problem, which can complement each other's advantages and overcome their respective shortcomings. Under certain assumptions, we have theoretically demonstrated the effectiveness of our method. Additionally, experimental results have shown that the method we proposed is a lightweight framework that outperforms previous methods of the same type on multiple types of noisy datasets.
>
>
> 5.
> We apologize for the oversight in our writing at the time. The links are highlight displayed during the editing process with the LaTeX compiler, lead to we neglect the generated PDF is difficult to read. We apologize for any inconvenience caused and will rectify this in the final version.
>
> Once again, we sincerely appreciate your patient review and suggestions. We hope that the additional explanations above can address your concerns and receive your approval.
>
> References
>
> [1] Zhou, Xiong, et al. "Learning with noisy labels via sparse regularization." Proceedings of the IEEE/CVF international conference on computer vision. 2021.
>
> [2] Liu, Sheng, et al. "Robust training under label noise by over-parameterization." International Conference on Machine Learning. PMLR, 2022.
>
> [3] Li, Xuefeng, et al. "Provably end-to-end label-noise learning without anchor points." International conference on machine learning. PMLR, 2021.

---

### Official Review · Reviewer_h1VJ · 2023-11-02

**Soundness:** 2 fair
**Presentation:** 3 good
**Contribution:** 2 fair
**Rating:** 5
**Confidence:** 4

**Summary:**

The paper introduces an enhanced transition matrix method for diverse label noise using implicit regularization. Traditional techniques for estimating per-sample transition matrices are computationally intensive and may be infeasible, particularly for instance-dependent noise. The proposed method estimates a single global matrix by gauging the probability transfer from accurate to noisy labels, adjusted via implicit regularization to bridge the gap between estimated posterior and noisy label distributions. This method, applicable to various noise types, addresses issues of posterior probability estimation inaccuracies. Supported by theoretical proofs and experimental results, the approach surpasses prior transition matrix techniques without necessitating auxiliary methods.

**Strengths:**

- The paper presents a novel approach to deal with label noise in machine learning models. It proposes the combination of global matrix estimation and implicit regularization to replace the cumbersome existing transition matrix methods. This is a creative combination of existing ideas, leveraging the strengths of each to propose a powerful, effective, and concise method for dealing with noisy labels in diverse situations.

- The experimental results reported in the paper demonstrate the proposed method's formidable performance, surpassing some robust algorithms based on sample selection and semi-supervised techniques.

**Weaknesses:**

-  The primary shortcoming of the paper lies in its lack of originality. While the idea of combining global noise transition matrix estimation with implicit regularization as an alternative to existing methods possesses some novelty, the concrete implementation, in my view, seems to be merely a straightforward amalgamation of SOP and VolMinNet. Furthermore, the derivations in the theorem-related sections bear considerable resemblance to those in SOP. Hence, from my perspective, the proposed algorithm appears more as an interpretation of SOP from an instance-dependent angle.

**Questions:**

Q1: The article claims that VolMinNet exhibits inaccuracies in its noisy posterior probability estimation, prompting the introduction of the method delineated in this paper as a solution. However, the experiments seem to overlook the state-of-the-art (SOTA) work, CCR, which also seeks to enhance VolMinNet. Would it be possible to include comparative experiments involving CCR?

- Cheng D, Ning Y, Wang N, et al. Class-Dependent Label-Noise Learning with Cycle-Consistency Regularization[J]. Advances in Neural Information Processing Systems, 2022, 35: 11104-11116.

Q2: The methodology section of the paper proposes estimating the discrepancy between \( P(\tilde{\boldsymbol{Y}} \mid X) \) and \( \boldsymbol{T}^{\top} P(\boldsymbol{Y} \mid X) \) using a feature-embedded regularization term. I'm curious, does this framework exhibit generality? In other words, would other regularization terms incorporating features produce similar or equivalent effects?

Q3: Indeed, there appears to be a potential inconsistency in the experimental setup. Comparing the results of a model using ResNet-18 as its backbone (TMR) with another using ResNet-34 (SOP) may not provide a fair comparison, especially given the capacity and potential performance differences between the two architectures. It becomes even more noteworthy if the TMR model with a ResNet-18 outperforms the SOP model with a ResNet-34 by a significant margin. This discrepancy might introduce biases in the evaluation and potentially affect the validity of the claims made.

---

> ### Author Response · Authors · 2023-11-15
>
> Dear reviewer,
>
> Thank you for your review of our paper and providing constructive questions and suggestions. We will address each of the questions you raised and are happy to continue further discussion on related issues with you.
>
> For Weaknesses:
>
> If only considering the form of the loss function, our method does look like a combination of VolMinNet and SOP. However, it is not simply a patchwork. Our core idea is to propose a relatively concise unified approach framework that can perform well on a wider range of noisy datasets. We are the first to propose the idea of combination transition matrix and implicit regularization to deal with the label noise problem, which can address the shortcomings of VolMinNet and SOP to obtain a model with a wider range of applicability. SOP can handle instance-dependent noise, but its performance is severely affected on datasets with high noise rates due to its requirement of noise sparsity. On the other hand, VolMinNet may have large errors when uses a fixed transition matrix to learn instance-dependent noise. Our proposed method (TMR) simultaneously mitigates the drawbacks of VolMinNet and SOP, resulting in a model with stronger fitting performance for more noisy datasets. Therefore, TMR is a meaningful job for label noise learning that can be considered as an extension of SOP for high noise rates, rather than an extension for instance-dependent angle.
>
> Furthermore, we provide detailed analysis in both theory and experiments to demonstrate the effectiveness of our proposed method, constituting a complete work. It is a simple and widely applicable model that can be used as a plug-and-play method. As for the doubt regarding the similarity between our theoretical analysis and SOP, it is true that we draw inspiration from SOP's method analysis and use some results in our proof process for simplicity, which are mentioned in paper. However, the models are different and we combine the analysis of transition matrix part. The construction and analysis process of our model are innovative and not just a naive move from SOP. The complexity and analysis difficulty of the model are higher than SOP. Besides, we do not think that the reference to previous method’s proof should be considered as a lack of originality. Moreover, we also give analysis on generalization in Section 3.2, which is not included in SOP.
>
> For Questions:
>
> For Q1:
>
> As you mentioned, CCR was proposed based on VolMinNet to alleviate the estimation error of noisy posterior probability. However, it is only applicable to class-dependent noise and does not work well for instance-dependent noise. Moreover, even for class-dependent noise, the framework of CCR is not as effective as our TMR, which can adjust the estimation of noisy posterior probability for each sample through implicit regularization. Taking the some results on CIFAR100 as an example, it can be observed that both for class-dependent noise data and instance-dependent noise data, there is a significant gap between CCR and our method.
>
> |  | Sym-20% | Sym-50% | Pair-20% | Pair-45% | IDN-20% | IDN-40% | 100N |
> |--------|--------|--------|--|--|--|--|--|
> | CCR | 67.74±0.17 | 59.73±0.63 | 71.63±0.39 | 69.18±1.30 | 68.33±0.43| 62.37±0.72 | 59.25±0.36|
> | TMR (Ours) | 75.90±0.23 | 70.93±0.44 | 76.03±0.20 | 70.86±0.56 | 75.92±0.44 | 72.09±0.70 | 64.28±0.27|
>
>
> For Q2:
>
> Our initial idea is to use a relatively small residual term $\gamma(X)$ to fit the difference between $ P(\tilde{\boldsymbol{Y}} \mid X)$ and $ \boldsymbol{T}^{\top} P(\boldsymbol{Y} \mid X)$, and then use $\boldsymbol{T}^{\top} P(\boldsymbol{Y} \mid X)+ \gamma(X)$ to fit the noisy labels distribution $ P(\tilde{\boldsymbol{Y}} \mid X)$. Choosing implicit regularization is based on its ability to achieve the effect of L1 regularization and it is easy to update and learn, inspired by method SOP. We have also tested to constrain L2 regularization and directly using L1 regularization in loss function before, but we failed to obtain the experimental results as good as the one mentioned in the paper.
>
> For Q3:
>
> In fact, we have experimented with ResNet34 on the CIFAR dataset and found that for our method, it performs slightly better than using ResNet18. However, our computing resources are limited. In order to simplify the experiments and reduce computational costs, we chose ResNet18. In terms of results, TMR with ResNet18 performs better than SOP with regardless of whether ResNet18 or ResNet34, especially as the noise rate increases. There should be no such concern for the Clothing1M and WebVision datasets, as both use the same underlying framework.
>
> Once again, we sincerely appreciate your patient review and suggestions. We hope that the additional explanations above can address your concerns and receive your approval.

---

### Author Response · Authors · 2023-11-15
**Further clarification on our work**

Dear reviewers,

We would like to express our gratitude to all reviewers for their diligent review and valuable suggestions. We have carefully addressed each of their comments in our responses. Throughout this process, we realized that some reviewers may have misunderstood our core ideas due to our failure to articulate them clearly in the paper. Therefore, we would like to provide a further clarification.

In statistics, there is a famous quote by George E. P. Box that states: "All models are wrong, but some are useful." Just take linear model as an example, they are widely used due to their simplicity in solving many problems, but they perform poorly in complex scenarios. Complex models may have better performance on individual tasks, but they lack generalization ability and suffer from excessive computational costs. In our work, we aim to find a relatively simple yet useful model that can handle various noisy label data and provide certain theoretical guarantees.

We construct our model by combining transition matrix and sparse implicit regularization, which enables statistical consistency under certain conditions without requiring additional complex network structures or excessive computational resources. While these two techniques have been individually explored in previous works, their combination in our approach is novel. The methodology section of our paper unfolds based on the construction principles of the model. We seek methods that can achieve the desired effects for each component, rather than simply assembling them to compose our work. We leverage the combination precisely because it can address the limitations of each other, allowing our model to achieve good results on a wider range of datasets. Ultimately, we obtain the desired TMR method.

Our theoretical analysis is based on the assumption that these conditions hold, leading to favorable theoretical results. In fact, the assumptions underlying the model constructed using transition matrix and implicit regularization may not hold for all datasets. However, experiments on a broader range of unknown noisy datasets demonstrate that even if our model assumptions are not perfect, our approach outperforms previous state-of-the-art transition matrix methods on numerous publicly available datasets. This indicates that our simplified model can effectively model various noisy datasets, making our method valuable in practical applications.

Furthermore, our method is designed to be a plug-and-play model that can be combined with any other advanced techniques, such as semi-supervised learning or contrastive learning. However, incorporating such techniques goes against our intention to develop a model that is both simple and universally applicable. Thus, in this paper, we did not compare our approach with these state-of-the-art methods but focused solely on comparing it with transition matrix-based methods, which have garnered attention due to their simplicity and statistical consistency.

We hope that these explanations will clarify our motivations and receive recognition from the reviewers. Thank you.

Best wishes,

The authors

---

### Meta-Review · Area_Chair_Ka3N · 2023-11-30

**Metareview:**

This paper worked on label-noise learning and proposed an improved way of noise transition matrix estimation --- a decomposition into a shared class-conditional part and an individual instance-dependent part --- this problem was claimed to be easier after the decomposition. The major issue is its limited novelty since the final proposal is a combination of mainly two existing methods, which was pointed out by all the 4 reviewers. Some other issues include over-claiming of the contributions and clarity of justifications. As a result, our reviewers consistently agreed to reject it.

**Justification For Why Not Higher Score:**

The major issue is its limited novelty since the final proposal is a combination of mainly two existing methods, which was pointed out by all the 4 reviewers. Some other issues include over-claiming of the contributions and clarity of justifications.

**Justification For Why Not Lower Score:**

N/A

---

### Decision · Program_Chairs · 2024-01-16

Reject